



# Impact of northern hemisphere mid-latitude anthropogenic sulfur dioxide emissions on local and remote tropospheric oxidants

Daniel M. Westervelt[1,2], Arlene M. Fiore[1,3], Colleen B. Baublitz[1,3], Gustavo Correa[1]

[1]Lamont-Doherty Earth Observatory, Columbia University. Palisades, New York, USA
[2]NASA Goddard Institute for Space Studies, New York, New York, USA
[3]Department of Earth and Environmental Sciences, Columbia University, Palisades, New York, USA

*Correspondence to*: Daniel M. Westervelt (danielmw@ldeo.columbia.edu)

**Abstract.** The unintended consequences of reductions in regional anthropogenic sulfur dioxide ($SO_2$) emissions implemented
to protect human health are poorly understood. $SO_2$ decreases began in the 1970s in the US and Europe and are expected to
continue into the future, while recent emissions decreases in China are also projected to continue. In addition to the well
documented climate effects (warming) from reducing aerosols, tropospheric oxidation is impacted via aerosol modification of
photolysis rates and radical sinks. Impacts on the hydroxyl radical and other trace constituents directly affect climate and air
quality metrics such as surface ozone levels. We use the Geophysical Fluid Dynamics Laboratory Atmospheric Model version
3 nudged towards National Centers for Environmental Prediction (NCEP) reanalysis wind velocities to estimate the impact of
$SO_2$ emissions from the United States, Europe, and China by differencing a control simulation with an otherwise identical
simulation in which 2015 anthropogenic $SO_2$ emissions are set to zero over one of the regions. Springtime sulfate aerosol
changes occur both locally to the emission region and also throughout the Northern Hemispheric troposphere, including remote
oceanic regions and the Arctic. The presence of sulfate aerosol strongly reduces hydroxyl (OH) and hydroperoxy ($HO_2$)
radicals by up to 10% year-round throughout most of the troposphere north of 30 ºN latitude, the latter of which is directly
removed via heterogeneous chemistry on aerosol surfaces, including sulfate, in the model. Regional $SO_2$ emissions
significantly increase nitrogen oxides ($NO_x$) by about 5-8% throughout most of the free troposphere in the Northern hemisphere
by increasing the $NO_x$ lifetime as the heterogeneous sink on sulfate aerosol declines. Despite the $NO_x$ increases, tropospheric
ozone decreases at northern mid-latitudes by 1-4% zonally averaged and by up to 5 ppbv in surface air over China, as its
response is dominated by the larger decreases (up to 2x) in $HO_2$ and OH. Since 2015 anthropogenic $SO_2$ emissions in China
exceed those in the US or Europe, the oxidative response is greatest for the China perturbation simulation. Chemical effects
of aerosols on oxidation (reactive uptake) dominate over radiative effects (photolysis rates), the latter of which are only
statistically significant locally for the large perturbation over China. We find that the emissions decrease in China, which has
yet to be fully realized, will have the largest impact on oxidants and related species in the Northern Hemisphere free
troposphere compared to changes in Europe or the USA. Our results bolster previous calls for a multipollutant strategy for air
pollution mitigation, to avoid the unintended consequence of aerosol removal leading to surface ozone increases that offset or



mask surface ozone gains achieved by regulation of other pollutants, especially in countries where current usage of high-sulfur emitting fuels may be phased out in the future.

## 1 Introduction

Understanding and constraining tropospheric oxidants such as the hydroxyl radical (OH) remains a key challenge of direct relevance to understanding the oxidizing power of the atmosphere, radiative forcing, and surface air quality. Despite the critical role in atmospheric chemistry, OH abundances differ widely among chemistry-climate and chemical transport models (Stevenson et al., 2020; Zhao et al., 2019). In addition, global, annual mean OH response to historical anthropogenic emission changes between the preindustrial and the present-day ranged from a 12.7% decrease to a 14.6% increase across 17 global models (Naik et al., 2013), with similar discrepancies across simulations of future composition and climate (Voulgarakis et al., 2013). These differences between model estimates of OH suggest major knowledge gaps in our understanding of the drivers of OH. One potential driver of tropospheric oxidant changes that has not received sufficient study is aerosols, which can uptake radical species (chemical effect) and scatter or absorb incoming solar radiation (radiative effect), thereby impacting OH and other important chemical species (Jacob, 2000; Wild et al., 2000).

Anthropogenic emissions of sulfur dioxide ($SO_2$), a precursor to sulfate aerosol, have significantly decreased in the United States and Europe for the last several decades and are projected to continue to decline (Riahi et al., 2011; Vuuren et al., 2011; Westervelt et al., 2015). In China, emissions of anthropogenic aerosols began to decline in about 2013 after increasing for decades (Fontes et al., 2017; Li et al., 2017; Samset et al., 2018). Previous research has indicated that these past and forthcoming emission changes have the potential to influence the tropospheric oxidation capacity on both a regional and global basis (Dentener and Crutzen, 1993; Dickerson et al., 1997; Martin et al., 2003). The aerosol decreases in China were associated with subsequent increases in summertime surface ozone ($O_3$) in China, attributed to a reduction in the sink of radical species such as the hydroperoxyl radical ($HO_2$) that promote $O_3$ production (Li et al., 2019b). Using a model and observations, the authors found that a 40% decrease in fine particulate matter ($PM_{2.5}$) in China between 2013 and 2017 led to an increasing ozone trend of up to 3 ppb per year in eastern China and was found to be a more important factor than $NO_x$ emissions reductions over the same time period (Li et al., 2019b, 2019a). These findings confirm earlier modelling work and point to an important role for aerosol impacts on tropospheric oxidation with implications for surface $O_3$ concentrations, especially over China (Li et al., 2018; Lou et al., 2014).

On a global scale, few studies have addressed the impact of aerosols on tropospheric oxidants. Often, aerosol impacts are assumed to be negligible in constraining present and future OH concentrations (Voulgarakis et al., 2013). Primary production of OH depends on the amount of water vapor and $O(^1D)$ present (formed via $O_3$ photolysis), and is the dominant pathway of OH formation in most locations except for high latitudes (Spivakovsky et al., 2000). Secondary production includes reactions of $HO_2$ or $RO_2$ (organic peroxy) radicals generated from oxidation of volatile organic compounds (VOCs) or carbon





monoxide (CO) with nitrogen oxide (NO) which regenerates OH. Concentrations of these atmospheric constituents and certain

meteorological factors such as absolute humidity, temperature, and ultraviolet radiation are thought to predominantly control OH abundance (Spivakovsky et al., 2000). However, by differencing a Goddard Earth Observing System Chemistry Transport Model (GEOS-Chem) control simulation of late 1990s atmospheric composition with a sensitivity simulation in which the offline global aerosols are excluded, Martin et al. (2003) find that the presence of all aerosols decreases OH by 9% globally and 5-35% in the northern hemisphere boundary layer. The authors also find 15-45 ppbv decreases in boundary layer $O_3$ over

India in March associated with the presence of all aerosols compared to all aerosols removed. In a similar global study, (Tie et al., 2005) use the Model for Ozone and Related Chemical Tracers version 2 (MOZART-2) to show that the net effect of all aerosols (natural and anthropogenic) reduces $HO_x$ (defined as OH + $HO_2$) and $O_3$ by 30% and 20%, respectively, improving on past methodology (e.g., Martin et al., 2003) by calculating aerosol abundances interactively. Past studies only considered global distributions of aerosols and often focused on natural aerosols such as dust or sea salt (Bian and Zender, 2003; Liao et

al., 2003). The impact of rapidly changing spatially heterogeneous anthropogenic aerosol abundances on tropospheric OH and $O_3$ is thus an open question.

We expand on past studies by considering individual regions (China, Europe, and the US) of sulfate aerosol decrease via $SO_2$ emissions reductions, and quantify the local and remote impacts of changing aerosol emissions on atmospheric $HO_x$, $NO_x$, and $O_3$ concentrations on a seasonal basis within a chemistry-climate model nudged to observed meteorology. We focus

on anthropogenic $SO_2$ emissions which have decreased most dramatically in many regions compared to anthropogenically-sourced carbonaceous aerosols or natural aerosols such as dust and sea salt. We seek mechanistic understanding on the interactions between aerosols, oxidants and radical species, and photolysis rates over different regions and in different seasons. We consider two main pathways through which aerosols can affect oxidation: modification of photolysis rates via extinction of incoming solar radiation (radiative effect) and heterogeneous uptake of radical species onto aerosol surfaces (chemical

effect). Finally, we consider the impact of anthropogenic $SO_2$ emissions reductions on boreal summertime surface $O_3$ concentrations in China, Europe, and the US.

## 2 Model and simulations

We use the National Oceanic and Atmospheric Administration Geophysical Fluid Dynamics Laboratory Atmospheric Model version 3 (GFDL-AM3), which is the atmosphere-only component of the GFDL coupled climate model, CM3 (Donner

et al., 2011). The model has been rigorously evaluated against observations in previous work (Donner et al., 2011; Naik et al., 2013). The model has 48 vertical layers from the surface up to about 0.01 hPa and a six-face cubed-sphere grid with 48 cells along each edge (C48), which is regridded to a 2 degree latitude by 2.5 degree longitude Cartesian grid. Emissions of anthropogenic trace gases and aerosols for year 2015 emissions are from the Representative Concentration Pathway 8.5 (RCP8.5) scenario (Riahi et al., 2011). The tropospheric chemical mechanism for aerosols and gas-phase species follows the

work of Horowitz et al. (2003) and Horowitz et al. (2007) with updates to photolysis, radical uptake by aerosols, and convective





wet scavenging of aerosols. The Fast-JX module (Bian et al., 2003; Wild et al., 2000) calculates the impact of online aerosols and clouds on photolysis rates and actinic fluxes, implemented into GFDL-AM3 according to Mao et al. (2013b). Heterogeneous uptake of radical species is simulated according to Mao et al. (2013b) and Mao et al. (2013a) using a first order reactive uptake rate constant $k$ (Equation 1):

$$k = -\left(\frac{r_e}{D_g} + \frac{4}{\gamma v}\right)^{-1} A \qquad\qquad\text{(Equation 1)}$$

where $r_e$ is the aerosol effective radius (m), $D_g$ is the gas-phase molecular diffusion coefficient, $v$ is the mean molecular speed of the gas, and $A$ is the aerosol surface area per unit volume of air. Here we set the heterogeneous reactive uptake coefficient ($\gamma$) of $HO_2$ to 0.2 instead of the value of 1.0 in Mao et al. (2013a). Though estimates of $\gamma$ are uncertain, recent literature suggests such high values of 1.0 are not supported by observations and that the parameter is likely closer to 0.2 (Abbatt et al., 2012; Li

et al., 2019a, 2019b; Taketani et al., 2012). Taketani et al.(2012) recommends a middle $\gamma$ value of 0.24 based on measurements at two high altitude sites in China. Reactive uptake coefficients for all other reactions including $N_2O_5$, $NO_3$, and $NO_2$ are shown in Table 1, taken from Jacob (2000). We allow uptake of $HO_2$, $N_2O_5$, $NO_3$, and $NO_2$ onto all aerosol types, including sulfate, black carbon, organic carbon, sea salt, and dust using the same coefficients for each composition. We also include updates to convective wet scavenging of aerosols in the form of finer vertical discretization of convective updraft plumes, resulting in

improvements in aerosol budgets (Paulot et al., 2015). Horizontal wind velocities are nudged using a pressure-dependent technique towards reanalysis values from the National Centers for Environmental Prediction Global Forecast System (NCEP GFS; (Lin et al., 2012)). Further model description and model evaluation against observations can be found in Donner et al. (2011), Naik et al. (2013a), and (Rasmussen et al., 2012).

We conduct a two-year (2014-2015) nudged control simulation in which emissions of aerosols and their precursors

follow RCP8.5 and contrast it with three perturbations: one in which all anthropogenic $SO_2$ emissions are set to zero over the United States (30ºN - 50ºN, 70ºW – 125ºW) , all anthropogenic $SO_2$ emissions are set to zero over Europe (35ºN – 70ºN, 15ºW – 55ºE), and all anthropogenic $SO_2$ emissions are set to zero over China (15ºN – 50ºN, 95ºE – 130ºE). $SO_2$ is oxidized by the hydroxyl radical in the gas phase and by ozone and hydrogen peroxide in clouds to form sulfate aerosol, which is a dominant component of total aerosol in GFDL-AM3 (Westervelt et al., 2015, 2017). We separately subtract each regional $SO_2$

perturbation simulation from the control simulation, thereby isolating the impact of $SO_2$ emissions (and subsequent sulfate formation) on tropospheric oxidants and related species. We test for statistical significance using a Student's t-test on seasonal mean responses with the null hypothesis being that the difference between the control and the perturbation simulation is zero. Only the full year of 2015 is used for analysis. $SO_2$ perturbations from our simulations are 10.8, 12.4, and 16.2 Tg $SO_2$ y$^{-1}$ for US, Europe, and China, respectively.

The global annual mean OH for the 2015 control simulation is 7.0 x $10^5$ molecules cm$^{-3}$, which is within the range of the 14 Atmospheric Chemistry and Climate Model Intercomparison Project (ACCMIP) for year 2000 and 14 Chemistry Climate Model Initiative (CCMI) models (Voulgarakis et al., 2013b; Zhao et al., 2019) for years 2000-2010. The global annual



tropospheric burden of $O_3$ in the 2015 control simulation is 356 Tg, which compares well to the year 2000 $O_3$ burden mean across the ACCMIP models of $337 \pm 23$ Tg (Young et al., 2013).

**3 Results**

Figure 1 shows the percent increase in seasonal (March-April-May, MAM) sulfate concentrations at the surface (right column) and at altitude (left column) for the presence of all anthropogenic US $SO_2$ (first row), all European $SO_2$ (second row), and all Chinese $SO_2$ (third row) based on year 2015 anthropogenic emissions. Additional seasons are shown in the supplemental (Figs. S1-3). The zeroing of 2015 emissions in each location results in the largest perturbation in China, where emissions are

highest, followed by Europe and the US. Sulfate increases are largest closest to the source region, but all three regional simulations show statistically significant remote impacts both horizontally and vertically in the atmosphere, as evidenced by the spatial and zonal plots in Fig. 1. The US, Europe, and China perturbations all significantly increase sulfate throughout the troposphere up to 200 hPa and higher towards the North Pole, with the largest increases of up to 30-40% resulting from a zeroing of China $SO_2$. Transport to the Arctic is a common feature in all three perturbations, and is consistent with previous

studies on aerosol transport to the Arctic (Shindell et al., 2008; Stohl, 2006). The US perturbation impacts sulfate concentrations significantly at the surface and at altitude over the North Atlantic Ocean, while the China perturbation exerts a heavy influence over the Pacific reaching all of the way to the Western US. European $SO_2$ emissions have widespread influence on the Northern Hemisphere, but especially in the Arctic and the Mediterranean and northern Africa. In all cases, sulfate changes are nearly entirely confined to the Northern Hemisphere.

We analyze the impact of sulfate changes on oxidation, starting in Fig. 2 with OH (left column) and $HO_2$ (right column) for each of the three regional perturbations (rows of Fig. 2). Sulfate aerosol surfaces directly uptake $HO_2$ radicals as described in Sect. 2, resulting in significant decreases of $HO_2$ and OH (via their rapid cycling). For each perturbation, decreases in both OH and $HO_2$ occur throughout most of the Northern Hemisphere up to about 200 hPa vertically during the boreal spring (MAM). The largest decreases in OH and $HO_2$ occur in spring for each of the perturbations, followed by winter (December-

January-February, DJF), autumn (September-October-November, SON), and summer (June-July-August, JJA). These additional seasons are plotted in Figs. S4-S6. In MAM, $SO_2$ emissions over the US decrease OH and $HO_2$ by about 5% within the US planetary boundary layer. In the mid-troposphere (400 – 600 hPa), OH decreases are 5% or greater and are located spatially above the Arctic. For the Europe $SO_2$ and China $SO_2$ cases during MAM, the Arctic middle troposphere OH decreases are larger in percent change (>10%) than the local changes near the surface (~8%). The presence of 2015 China $SO_2$ emissions

also decreases OH and $HO_2$ by about 10% over the north Pacific Ocean middle troposphere (about 400-600 hPa). We conclude that regional $SO_2$ emissions may have stronger impacts remotely than locally, and OH may be relatively more sensitive to aerosol changes in the Arctic and remote oceans at higher altitudes where its production is more limited.

In Fig. 3 we plot spring (MAM) changes in $NO_x$ (defined as $NO + NO_2$) concentrations in response to anthropogenic $SO_2$ emissions in the US, Europe, and China. While $HO_2$ and OH strongly decreased in response to $SO_2$ emissions, $NO_x$



significantly increases throughout most of the Northern Hemisphere. In the model, aerosols can take up $NO_2$ directly but with a very low reaction probability (0.0001, Table 1), such that little uptake actually occurs and is easily offset by feedbacks onto other chemical reactions involving $NO_x$. Instead, reduction in the sinks of $NO_x$ via OH (nitric acid formation) during the day and uptake of $NO_x$ reservoir species at night dominates the response to $SO_2$ emission changes, increasing $NO_x$ in the model as OH decreases. At night, $NO_x$ is removed by reaction with the nitrate radical ($NO_3$), which forms dinitrogen pentoxide ($N_2O_5$)

(Chang et al., 2011; Jacob, 2000). Sulfate aerosols are effective at removing both $NO_3$ and $N_2O_5$ via reactive uptake (reaction probability of 0.1), slowing down these night time $NO_x$ sinks and thus increasing $NO_x$ abundance. This hindering of $NO_x$ sinks is most effective during MAM and DJF in the Northern Hemisphere mid-troposphere (Fig. 3a-c for MAM, additional seasons shown in Figs. S7-S9). Mid-tropospheric northern hemisphere $NO_x$ increases reach about 7-8% in response to Chinese $SO_2$ emissions specifically, with smaller effects for both US and Europe $SO_2$ perturbations. $NO_x$ at the surface increases slightly

less at about 5-7% depending on the regional emissions perturbation, though these changes still are statistically significant. Several previous studies have used smaller reactive uptake coefficients for $N_2O_5$ than the value of 0.1 used here (Evans and Jacob, 2005; Holmes et al., 2019; Macintyre and Evans, 2010; McDuffie et al., 2019) based on more recent laboratory experiments, but only find impacts on mean tropospheric $O_3$ burden of 2-4%. Using a box modelling approach, McDuffie et al. (2019) find a median $\gamma$ for $N_2O_5$, of 0.076, reasonably close to our assumed value of 0.1. Using a smaller $\gamma$ would lead to

less $N_2O_5$ uptake by aerosols, a smaller decrease in the $NO_x$ sinks, and therefore a smaller increase in $NO_x$ concentration via the $N_2O_5$ uptake pathway.

In most of the Northern Hemisphere troposphere, $O_3$ decreases in MAM by up to 4% in response to US, European, or Chinese $SO_2$ emissions increases in the model (Fig. 4), mostly coinciding with regions of large $HO_x$ decreases (Fig. 2), despite the increase in $NO_x$ (Fig. 3). The $O_3$ increases in the upper troposphere are mostly not significant. We examine model

diagnostics of gross ozone production (the sum of $HO_2$+NO and all $RO_2$+NO reaction pathways) and $O_3$ loss (which includes reaction of $O_3$ with $HO_x$ and with alkenes, plus $O_3$ photolysis followed by $O^1D$+$H_2O$) to interpret further the $O_3$ decrease. While both $O_3$ production ($P_{O3}$) and loss ($L_{O3}$) rates decline (Fig. S10 and Fig. S11), production decreases more strongly than loss, lowering $O_3$ concentrations. We confirm that transport of $O_3$ from other latitudes is unlikely to contribute much to the modelled $O_3$ response as the change in advective or convective tendency in $O_3$ (Fig. S12) is far smaller than the chemical

production and loss terms (Fig. S10). The $O_3$ production and loss rates decrease most strongly in the lower troposphere over the source regions (Fig. S10 and Fig. S11) while the $O_3$ decreases (Fig. 4) propagate more widely through the free troposphere, indicating reduced export from these source regions. Additional seasons for $O_3$ change are shown in Figs. S13-S15.

We find here that the decline in $HO_x$ and its impact on $P_{O3}$ outweighs the aerosol-induced increases in $NO_x$ and decreases in $O_3$-$HO_x$ sinks, even during summer in all three source regions. We show the response of summertime surface 8-

hour maximum daily average (MDA) $O_3$ to increasing anthropogenic $SO_2$ emissions in the US, Europe, and China in Fig. 5. Increasing sulfate aerosol increases the sink of $HO_2$ radicals and thus slows down $O_3$ production (Fig. S11), resulting in surface $O_3$ concentration decreases. Sulfate aerosol can also reduce $NO_2$ and $O_3$ photolysis rates. The combined effect of sulfate aerosol on changes in photolysis rates and heterogeneous chemistry is a statistically significant decrease of about 5 ppbv over most of





eastern China, Korea, and Japan when Chinese $SO_2$ emissions are introduced, a decrease of about 3 ppbv over the eastern US

for US $SO_2$ emissions, and a decrease of about 3 ppbv over Eurasia for the Europe $SO_2$ perturbation. Decreases in surface MDA8 $O_3$ are mostly confined to near the source region. Changes in similar magnitude have been reported over China using both a chemistry-transport model and observations (Li et al., 2019). Large sulfate decreases have occurred since the 1970s in both Europe and the US. The $SO_2$ perturbation in our study (zero-out 2015 level emissions) is 10.8, 12.4, and 16.2 Tg $SO_2$ $y^{-1}$ in the US, Europe, and China, respectively. These results imply that the sulfate decreases from clean air regulations and

technologies have had the unintended consequence of driving $O_3$ up by a few ppb during the summertime in the US and Europe. $NO_x$ emissions have also decreased dramatically over roughly the same time period and have likely more than offset any $O_3$ increase from decreasing sulfate. However, the full potential of possible $O_3$ improvement via $NO_x$ and anthropogenic volatile organic carbon (VOC) decreases may have been partially masked by sulfate decreases. These findings highlight the importance of a multi-pollutant strategy for effective clean air regulation.

Finally, in Fig. 6 we plot the relative change in MAM O3→O($^1$D) and $NO_2$ photolysis rates, denoted $j_{o1D}$ and $j_{NO2}$ in response to $SO_2$ emissions in each region. Photolysis of both species is slightly influenced by changing $SO_2$ emissions, especially over China in response to China $SO_2$ emissions, where decreases in both photolysis rates are about 7%. For each of the perturbations, especially the US and Europe cases, changes in photolysis rates rarely rise above the noise, which is likely caused by meteorological factors such as slight changes in cloud cover. We conclude that while radiative effects via photolysis

are non-negligible, they are significantly less important than chemical effects for aerosol impacts on oxidation, consistent with previous findings (Li et al., 2019).

## 4 Summary and conclusions

Using the updated GFDL-AM3 nudged chemistry-climate model with online aerosol heterogeneous chemistry and interactions with radiation, we estimate the impact of northern hemisphere mid-latitude regional anthropogenic $SO_2$ emissions

on tropospheric OH, $HO_2$, $O_3$, and $NO_x$. Regional $SO_2$ emissions perturbations lead to significant changes to sulfate aerosol in far-reaching regions of the world, particularly in the Arctic and the mid and upper troposphere. OH and $HO_2$ decrease throughout the northern hemisphere mid-troposphere by up to 10%, which in turn increase $NO_x$ concentrations by at least 5%. $NO_x$ is not efficiently removed by heterogeneous reactions on aerosols, while species that contribute to $NO_x$ sinks such as OH (via $HO_2$ uptake), $N_2O_5$, and $NO_3$ are efficiently removed, slowing down the $NO_x$ sink and increasing $NO_x$ concentrations.

However, any influence of $NO_x$ increases on tropospheric $O_3$ are overwhelmed by $HO_2$ decreases, and the resulting decrease in $O_3$ production offsets decreases in $O_3$ sinks, resulting in up to 4% decrease in $O_3$ in the free troposphere and at the surface. Aerosols impact oxidation primarily through heterogeneous reactive uptake pathways over photolysis pathways.

Surface ozone decreases by 3 to 5 ppbv in response to the introduction of regional $SO_2$ emissions. If $SO_2$ emissions decline in developing regions of the world such as South Asia and sub-Saharan Africa, a goal attained through air quality

improvements to protect human health, there could be an unintended increase in surface $O_3$ concentrations. Decreasing surface



O$_3$ in these regions will require a multipollutant approach in which NO$_x$ and VOCs are simultaneously decreased with aerosols in order to offset the effect of decrease in aerosols and their precursors. While SO$_2$ and NO$_x$ emissions decreases coincided to some extent in the US, end-of-pipe technologies at power plants allow for control of SO$_2$ and NO$_x$ individually, and other sources of fine particulate matter (PM$_{2.5}$) such as waste burning and vehicle emissions will have a similar effect on ozone as

sulfate aerosols. PM$_{2.5}$ and SO$_2$ have decreased dramatically in recent years in the US and Europe, such that O$_3$ improvements may have been partially masked by the aerosol impact. SO$_2$ perturbations from our simulations are 10.8, 12.4, and 16.2 Tg y$^-$ $^1$ for US, Europe, and China, respectively, which result in a 3 to 5 ppbv surface ozone response over China, where SO2 emission are the largest in 2015.

Future work is needed to improve estimates of reactive uptake of HO$_2$ and other radical species by aerosols, as great

uncertainty still exists surrounding this parameter as well as the dependence of aerosol composition on reactive uptake parameters (George et al., 2013). We focus here on anthropogenic aerosols as they are changing rapidly and expected to continue to change. Previous work finds a large influence of Saharan dust aerosols on oxidation (Tie et al., 2005). Regions of biomass burning such as Africa and South America are also potential contributors to aerosol-driven oxidation changes. In order to avoid "trading one problem for another" in areas of the world that are experiencing both rapid emissions changes and high

exposures to air pollutants, we must better understand the impact of aerosols on atmospheric photochemistry.

**Author Contributions**

DMW wrote the manuscript, created all figures, and conducted all simulations. AMF and DMW originally conceived the project. CBB assisted with model setup and analysis of output. GC developed the model for use at LDEO. All authors

contributed to editing the manuscript.

**Code Availability**

The code for GFDL-AM3 is available here: https://www.gfdl.noaa.gov/am3/.

**Data Availability**

Data is available here: https://figshare.com/articles/dataset/Concentration_data_for_aerosol_impact_on_oxidants/13331066

(Westervelt, 2020)

**Competing Interests**

The authors declare no competing interests.

**Acknowledgements**



The authors acknowledge funding from the NASA Atmospheric Composition, Modeling, and Analysis Program (ACMAP)
grant number NNX17AG40G. We thank Drs. Bryan Duncan and Melanie Follette-Cook of NASA GSFC for their helpful
conversations.

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

**Table 1: Heterogeneous reactive uptake coefficients for several reactions in GFDL-AM3**

| Reaction | Uptake coefficient ($\gamma$) |
| --- | --- |
| $HO_2 \rightarrow H_2O_2$ or $H_2O$ | 0.2 |
| $N_2O_5 \rightarrow 2.0\ HNO_3$ | 0.1 |
| $NO_3 \rightarrow 1.0\ HNO_3$ | 0.1 |
| $NO_2 \rightarrow 0.5\ HNO_3$ | 0.0001 |






**Figure 1: Boreal springtime (MAM) mean percent change in sulfate concentration between a control simulation and a perturbation simulation in which anthropogenic SO2 emissions are removed over a certain region: (a,b) US, (c,d) Europe, and (e,f) China. Hatching denotes statistical significance according to a Student's t-test at the 95% confidence level.**







**Figure 2: Boreal springtime (MAM) mean percent change in OH (left column) and HO$_2$ (right column) between a control simulation and a perturbation simulation in which anthropogenic SO2 emissions are removed over a certain region: (a,b) US, (c,d) Europe, and (e,f) China. Hatching denotes statistical significance according to a Student's t-test at the 95% confidence level.**







**Figure 3: Boreal springtime (MAM) mean percent change in NOx between a control simulation and a perturbation simulation in which anthropogenic SO2 emissions are removed over a certain region: (a) US, (b) Europe, and (c) China. Hatching denotes statistical significance according to a Student's t-test at the 95% confidence level.**






**Figure 4: Boreal springtime (MAM) mean percent change in O3 between a control simulation and a perturbation simulation in which anthropogenic SO2 emissions are removed over a certain region: (a) US, (b) Europe, and (c) China. Hatching denotes statistical significance according to a Student's t-test at the 95% confidence level.**






**Figure 5: Summertime (JJA) surface O3 change (in ppbv) between a control simulation and a perturbation simulation in which anthropogenic SO2 emissions are removed over a certain region: (a) US, (b) Europe, and (c) China. Hatching denotes statistical significance according to a Student's t-test at the 95% confidence level.**






**Figure 6: Boreal springtime (MAM) mean percent change in photolysis rates ($j_{o1d}$, left column and $j_{no2}$, right column) between a control simulation and a perturbation simulation in which anthropogenic SO2 emissions are removed over a certain region: (a,b) US, (c,d) Europe, and (e,f) China. Hatching denotes statistical significance according to a Student's t-test at the 95% confidence level.**