# Peer review of "Impact of regional northern hemisphere mid-latitude anthropogenic sulfur dioxide emissions on local and remote tropospheric oxidants"

_Atmospheric Chemistry and Physics, 2020_

## Author Comment (AC1)

Below we respond to referee comments. Referee comments are in blue color text, while our responses are in normal black color text.

Referee 1:

This study examined the impacts of SO2 emissions from the United States, Europe, and China on tropospheric oxidants and ozone concentrations using GFDL-AM3 nudged towards NCEP wind velocities. They found that OH and HO2 decrease throughout the northern hemisphere mid-troposphere by up to 10%, as well as up to 4% decrease in O3 in the free troposphere and at the surface due to the presence of SO2. They also indicated that aerosols impact oxidation primarily through heterogeneous reactive uptake pathways over photolysis pathways. They topic is interesting and within the scope of the journal. The manuscript is well organized. Before it can be considered to be accepted, a few comments shoud be addressed.

We thank the reviewer for their comments.

General comments: This study investigates the responses of tropospheric oxidants to SO2 emissions from the United States, Europe, and China. In this case, I would like to see what are the response differences between these three regional sources. The authors only mentioned the oxidative response is greatest for the China perturbation simulation due to this largest emission. More detailed analysis/information should be given.

Each figure already shows the oxidative response for each of the three regional cases (US, Europe, and China), in separate rows. The response differences are given, for example, by comparing panels (a) and (b) to panels (e) and (f) in Figure 2. There are also more similarities than differences in the oxidative response to anthropogenic SO2 emissions each of the three regions, at least zonally. To use Figure 2 as an example again, the OH and HO2 responses seem to have a very similar zonal mean structure, with the key difference being the larger magnitude of the change correlating with the larger emissions perturbation. At the surface, the responses are more different, as can be seen in Figure 1, and also in Figure S11. There is already quite a bit of comparison of the different regions for SO4 concentration, but less so for the oxidative response. We therefore add the following to the manuscript:

Section 3, 2$^{nd}$ paragraph:

"By comparing the first row of Fig. 2 with the second and third rows, we find that the zonal structure of the OH and HO$_2$ response to anthropogenic SO$_2$ emissions is very similar across the three regional perturbations, while the magnitude is largest in response to China SO$_2$ emissions, followed by Europe SO$_2$, and US SO$_2$."

Section 3, 4$^{th}$ paragraph, referring to Figure 4 and ozone changes:

"O$_3$ decreases are the largest in response to Chinese anthropogenic SO$_2$ emissions, owing to the larger SO$_2$ perturbation compared to the US and Europe emissions perturbations."

Section 3, 5$^{th}$ paragraph, referring to Figure 5 and the P(O3) and L(O3):

"Increasing sulfate aerosol increases the sink of $HO_2$ radicals and thus slows down $O_3$ production (Fig. S11), resulting in surface $O_3$ concentration decreases, which are largest and mostly confined to the emissions source region."

We have also modified one sentence in the conclusions to give the reader a better sense of the difference between the various perturbations:

"$SO_2$ perturbations from our simulations are 10.8, 12.4, and 16.2 Tg y$^{-1}$ for US, Europe, and China, respectively, which result in a 3 **ppbv surface ozone response over the US and Europe, and a 5 ppbv surface ozone response over China**, where $SO_2$ emissions are the largest in 2015."

A comparison based on Fig. 5 (surface ozone) was already written into the original manuscript:

"The combined effect of sulfate aerosol on changes in photolysis rates and heterogeneous chemistry is a statistically significant decrease of about 5 ppbv over most of eastern China, Korea, and Japan when Chinese $SO_2$ emissions are introduced, a decrease of about 3 ppbv over the eastern US for US $SO_2$ emissions, and a decrease of about 3 ppbv over Eurasia for the Europe $SO_2$ perturbation."

Specific comments:

Title: This study focused on the SO2 emissions from the United States, Europe, and China instead of the whole northern hemisphere mid-latitude anthropogenic sulfur dioxide emissions. I would suggest to make it more specific.

We have added the word "regional" in the title to indicate that it is not the whole northern hemisphere and rather individual regions of the northern hemisphere. New title:

Impact of regional northern hemisphere mid-latitude anthropogenic sulfur dioxide emissions on local and remote tropospheric oxidants

Lines 19-21: This study zeroed out the regional emissions individually. But this sentence is more like all emissions were zeroed out.

Added the phrase "in three individual regions" to clarify.

"The presence of sulfate aerosol **in three individual regions** strongly reduces hydroxyl (OH) and hydroperoxy ($HO_2$) radicals by up to 10% year-round throughout most of the troposphere north of 30 ºN latitude, the latter of which is directly removed via heterogeneous chemistry on aerosol surfaces, including sulfate, in the model."

Line 40: The OH response to historical anthropogenic emissions renged from a 12.7% decrease to a 14.6%. Which anthropogenic emissions are inferred here, aerosol, GHGs, ozone?

Aerosols, GHGs, and ozone precursors. Essentially all anthropogenic emissions. GHGs are actually input to the model as concentrations rather than emissions in the paper that is being referred to. The comparison is from preindustrial (1850) to present-day (2000). We have added "(all species)" to the sentence to clarify.

"In addition, global, annual mean OH response to historical anthropogenic emission changes **(all species)** between the preindustrial and the present-day ranged from a 12.7% decrease to a 14.6% increase across 17 global models (Naik et al., 2013b),"

Line 65: The authors mentioned absolute humidity, temperature, and ultraviolet radiation are thought to predominantly control OH abundance. Aerosol change can also influence humidity and temperature. Why they were not disscussed?

While it is of course true that aerosols can influence humidity and temperature, aerosol impacts on climate cannot be easily diagnosed in a nudged climate model, as the nudging configuration will override any possible (minimal) aerosol impacts.

Line 90: Which aspects of the model were evaluated against observations? Since this study focused on the impacts of SO2 on HOx, NOx, and O3, at least, SO2, sulfate and ozone concentrations over China, US and Europe shoud be evaluated.

$O_3$ model evaluation has already published several times for a very similar version of the model, such that a full evaluation here would be superfluous. See for example, Figure 4 of Naik et al. (2013b) for ozone over the US and Europe, which we already cite. Surface ozone was evaluated thoroughly over China in Westervelt et al. (2019), which we had not cited in the original manuscript but do now. See in detail Figure S3 of that paper. Together, those two papers cover GFDL-AM3 model evaluation for surface $O_3$. Sulfate has been evaluated over the US and Europe in Paulot et al., (2015). They find a normalized mean bias against surface sulfate concentrations of 0.07 over the US and -0.43 over Europe, indicating good agreement over the USA, and moderate agreement over Europe. We do not have access to surface sulfate observations over China, but note that the studies cited in the manuscript evaluate aerosol optical depth (AOD) over China which can give a rough idea of the model performance for sulfate, a major contributor to overall AOD. Given the previous evaluation of GFDL-AM3 against observations, we do not repeat this previously published analysis.

We have expanded the section about model evaluation:

"The model has been rigorously evaluated against observations in previous work, **including against surface observations of $O_3$ over the US, Europe, and China (Donner et al., 2011; Naik et al., 2013a, Westervelt et al., 2019). Paulot et al. (2016) evaluate sulfate concentrations in GFDL-AM3 over the US (Interagency Monitoring of Protected Visual Environments, IMPROVE) and Europe (European Monitoring and Evaluation Programme, EMEP), and find a normalized mean bias of 0.07 in model surface concentrations compared against IMPROVE and a -0.43 mean normalized bias against EMEP."**

We have also added the following line to the conclusions to better contextualize our results:

"Model overprediction of surface $O_3$ over urban areas in China (Westervelt et al., 2019) likely make this 5 ppbv change an upper estimate of the surface $O_3$ response to China $SO_2$ emissions."

Line 121: How the authors test the significance of the difference with only one year simulation? The test requires two arrays. With one year simulation for both ctrl and sensitivity simulations, I am not sure how these two arrays were created.

It is one year each for two separate simulations. One array is the one-year time series of the response variable (e.g., OH) in the perturbation simulation (e.g. zero US anthropogenic SO2) and the other array is the response variable in the control simulation. The t-test determines whether the difference between these two simulations is statistically significantly different from zero.

Line 134: The figures only show the relative change in percentage. I do not see the largest perturbation in China, although it shoud be true in absolute concentration change.

The word "relative" has been added to this line:

"The zeroing of 2015 emissions in each location results in the largest **relative** perturbation in China, where emissions are highest, followed by Europe and the US."

Lines 139-144: Many recent studies also analyzed the long-range transport of aerosols between continents and to the Arctic (e.g., Yang et al., 2017, 2018; Ren et al., 2020). The authors may consider to cite more relavent studies.

 All three have been cited.

References:

Yang, Y., Wang, H., Smith, S. J., Easter, R., Ma, P.-L., Qian, Y., Yu, H., Li, C., and Rasch, P. J.: Global source attribution of sul- fate concentration and direct and indirect radiative forcing, Atmos. Chem. Phys., 17, 8903–8922, https://doi.org/10.5194/acp- 17-8903-2017, 2017.

Yang, Y., Wang, H., Smith, S. J., Easter, R. C., and Rasch, P. J.: Sulfate aerosol in the Arctic: Source attribution and radiative forcing, J. Geophys. Res., 123, 1899–1918, https://doi.org/10.1002/2017JD027298, 2018.

Ren, L., Yang, Y., Wang, H., Zhang, R., Wang, P., and Liao, H.: Source attribution of Arctic black carbon and sulfate aerosols and associated Arctic surface warming during 1980–2018, Atmos. Chem. Phys., 20, 9067–9085, https://doi.org/10.5194/acp-20-9067-2020, 2020.

**Referee 2**

This study documents the impact of anthropogenic SO2 emissions from China, Europe and the US , on tropospheric oxidants in the GFDL AM3 model. The authors test this by conducting individual simulations where the SO2 emissions in each region are systematically switched off. They find SO2 emissions and subsequent formation of sulphate aerosols lead to a decrease in OH and HO2, increase in NOx and a decrease ozone. Changes in concentrations are focussed predominantly in the northern hemisphere. The chemical effect, via heterogenous uptake of these species by aerosol particles dominates over the photolysis effect , whereby aerosol alter the photolysis rates of O3 and NO2. This is a pertinent and timely study given current strategies to improve air quality, particularly in Asia and ensuring all atmospheric chemistry aspects and pollutants impacting this are considered. It is a well written piece of work and clearly laid out.

I therefore find it is suitable for publication in ACP subject to some minor comments which should be addresses prior to publication.

We thank the reviewer for their comments.

Minor comments:

- Does the model include nitrate, and if so how does this impact sulfate formation and subsequent impact on oxidants ?

Nitrate aerosols are included in this version of the model. Nitrate can play a role in heterogenous uptake, and the same uptake coefficient for a given species (e.g. $HO_2$) is applied to all aerosol species (this is already mentioned in the original manuscript). Nitrate chemistry is implemented in a rather simple manner, in which ammonium nitrate formation is calculated following Stelson and Seinfeld (1982) but the reaction is considered irreversible. Nitrate aerosol formation in this setup will have minimal impacts on sulfate formation, as ammonium sulfate formation will dominate except for very ammonia-rich environments. Additionally, the majority of the $SO_2$ oxidation in the model occurs in-cloud as opposed to gas-phase, where competition with nitrate is less important. We paste a figure below which shows a near zero ammonium nitrate response to the reduction in $SO_2$, indicating that nitrate is not playing much of a role in the ammonium sulfate formation.

[Figure]

**Figure: Boreal springtime (MAM) mean percent change in ammonium nitrate concentration between a control simulation and a perturbation simulation in which anthropogenic SO2 emissions are removed over a certain region: (a,b) US, (c,d) Europe, and (e,f) China. Hatching denotes statistical significance according to a Student's t-test at the 95% confidence level.**

Ref:
Stelson, A. W. and Seinfeld, J. H.: Relative humidity and temperature dependence of the ammonium nitrate dissociation constant, Atmos. Environ., 16, 983–992, doi:10.1016/0004-6981(82)90184-6, 1982.

- The changes in surface ozone (of the order of 3ppb/year) , this would equate to less than 10% or so of the surface ozone in the source regions studied here and is likely well within the model standard deviation. Can the authors comment on the role of the model uncertainty /variability on the significance of these results and potential implications for AQ. Indeed, in L199/200 the authors refers to clean air technologies driving up O3 by "a few ppb" in summertime. A comment on the relative significance of these changes for local air quality should be provided.

The largest changes over China (e.g. Fig. 5c) are >5 ppbv. Additionally, in all regions, changes are statistically significant at the 95% confidence level, meaning they are departures from the mean of at least 2 times the standard error. We therefore disagree that the changes are within the model standard deviation, though the larger point about uncertainty and variability is still valid. After the sentence referred to by the referee, we add the following:

"While this may be a small amount of the total surface $O_3$ concentration and not entirely outside the range of typical variability, our study only considers the impact of sulfate aerosol and not carbonaceous aerosols, which make up greater than 50% of the total aerosol mass in many environments (Jiminez et al., 2009). Additionally, even O3 changes on the order of 3-5 ppbv may be important for holistically meeting tightening air quality standards.

- L147: how much of the reduction in OH is attributed to the heterogenous uptake of sulphate by HO2 versus increase sink via oxidation of SO2+OH? Can the authors separate out these contributions to OH loss. The analysis presented focus on the radiative effect (via changes in photolysis) and the chemical effect ( heterogeneous uptake) but what about the additional aerosol oxidation chemistry that is also taking place and how that impacts the oxidant concentrations?

This is an interesting thought that we unfortunately do not have the ability to diagnose. However, given the relatively long timescale of the $SO_2$ + OH reaction (~20 hours) compared to the short lifetime of the $HO_2$ radical, we expect that $HO_2$ uptake and fast cycling with OH is the dominant loss pathway.

Technical comments:

L21: The sentence "Regional SO2 emissions ….. increasing the NOx lifetime as the heterogeneous sink on sulfate aerosol declines" is confusing as the sink on sulfate of OH, NO3 and N2O5 are increased which leads to the reduction in NOx sinks. Please rephrase.

Phrased as "heterogeneous sink of $HO_2$ on sulfate" to clarify which sink we are talking about.

L132: for the presence of --> due to the presence of

Done

L134: The zeroing of 2015 emissions --> the zeroing of 2015 **SO2** emissions

Done

L138/9: from a zeroing of China SO2 --> the increase in sulphate clearly doesn't come for the zeroing of the emissions, suggest the statement is rephrased.

Changed to "from the China $SO_2$ perturbation" to fix.